# European Brown Hare Syndrome in Poland: Current Epidemiological Situation

**DOI:** 10.3390/v14112423

**Published:** 2022-10-31

**Authors:** Andrzej Fitzner, Wiesław Niedbalski, Andrzej Kęsy, Bogusław Rataj, Marian Flis

**Affiliations:** 1Department of Foot and Mouth Disease, National Veterinary Research Institute, ul. Wodna 7, 98–220 Zduńska Wola, Poland; 2Polish Hunting Association, District Board Nowy Sącz, 33–300 Nowy Sącz, Poland; 3Department of Animal Ethology and Wildlife Management, Faculty of Animal Sciences and Bioeconomy, University of Life Sciences in Lublin, ul. Akademicka 15, 20–950 Lublin, Poland

**Keywords:** EBHS, European hare, lagoviruses, serology, epidemiology

## Abstract

European brown hare syndrome (EBHS) is one of the main causes of mortality in brown hares (*Lepus europaeus*) and mountain hares (*Lepus timidus*) in Europe. Since the mid-1990s, this highly lethal and contagious plague has been widespread in many European countries, contributing to a drastic decline in the number of free-living and farmed hares. A second lagovirus, able to infect some species of hares is rabbit haemorrhagic disease virus 2 (RHDV2; GI.2) recognised in 2010, a new viral emergence of RHDV (GI.1) which is known to be responsible for haemorrhagic disease in rabbits—RHD. The aim of this study was to evaluate the current EBHS epidemiological situation on the basis of the presence of antibodies to European brown hare syndrome virus (EBHSV) and anti-RHDV2 antibodies in sera collected from free-ranging hares in Central and Southeastern Poland in 2020–2021. Additionally, studies on the presence of EBHSV and RHDV2 antigens or their genetic material in the blood and internal organs taken from brown hares between 2014–2021 have been carried out. The results of the serological examination showed nearly 88% of tested blood samples were positive for EBHSV antibodies. No EBHSV was identified in the examined hares using virological and molecular tests. The positive results of EBHS serological studies confirmed the circulation and maintenance of EBHSV in free-living brown hares in Poland. However, no serological, virological or molecular evidence was obtained indicating that the brown hares tested had been in contact with RHDV2.

## 1. Introduction

European brown hare syndrome (EBHS) is a highly contagious and fatal viral disease of the European brown hare (*Lepus europaeus*) and mountain hare (*Lepus timidus*) affecting free-ranging and farmed hares in Europe. The disease was first recognised in 1980 in Sweden, although there are many indications of its earlier occurrence [1,2,3,4,5,6]. In the mid-1990s, as a result of the intensive spread of the highly lethal and contagious EBHSV in many European countries, a drastic decline in the number of free-living hare populations was noted [7,8,9,10,11,12,13,14]. EBHS shares clinical, pathological and epizootic similarities with another fatal viral disease occurring in Leporidae—rabbit haemorrhagic disease (RHD)—recognised in 1984 [15]. Both aetiological agents, EBHSV (GII.1) and RHDV (GI.1–2) are members of the family *Caliciviridae*, genus *Lagovirus* [16]. The first EBHSV isolate in Poland was confirmed in 1992 [17,18]. At that time, the presence of EBHS infections in the native hare population was also proven on the basis of serological investigations [19,20].

Diagnostic methods of EBHS include serological tests, such as ELISA tests using monoclonal antibodies (e.g., IZSLER kits, Brescia, Italy) to detect the presence of EBHSV-specific antibodies, as well as virological tests for direct recognition of the antigen formed by the major structural protein of the capsid (VP60). Both conventional and real-time RT-PCR tests are available for the detection of EBHSV RNA [21,22].

In addition to the natural host, EBHSV genetic material was also detected in faecal swabs of wolves [23] and in fox carcasses [24], indicating that sympatric hare predators may play a role in the passive dissemination of the EBHSV.

In 2010, a new type of RHDV (GI.1), RHDV2 (GI.2) was diagnosed in France [25]. RHDV2, similar to RHDV and its antigenic subtype RHDVa (GI.1a) is responsible for a fatal infection in wild and domesticated rabbits (*Oryctolagus cuniculus*). However, RHDV2, unlike to the previous two forms of this pathogen, is also capable of overcoming the interspecies barrier and causing a fatal disease with clinical symptoms similar to EBHS in various hare species, including brown hares as well as infecting non-lagomorph species [22,26,27,28,29,30,31,32]. The first Polish RHDV2 outbreak was confirmed in 2016 in Łódź Voivodeship [33]. Recently, the presence of lagoviruses recombinants of the non-structural part of the EBHSV genome and structural genes of RHDV2, resulting in the origin of a novel form of RHDV2, was found [34]. At present, only one serotype of EBHSV is known, whereas due to the significant antigenic differences found between RHDV2 and RHDV/RHDVa, two serotypes of RHD causative lagoviruses can be identified [25,35,36].

The purpose of this study was to evaluate the current EBHS situation in Poland based on a serological survey as well as virological and molecular analysis of samples obtained from a free-living population of brown hares. The main serological study was carried out on the native brown hares hunted in Central and Southeast Poland, for two winter seasons, from December 2020 to December 2021. The blood samples and organ specimens were taken from hares of both sexes, adults and young, of different weights in hunting districts managed by Polish Hunting Association. Moreover, the results of the study on the presence of RHDV2 in native, free-living hares by the serological, virological and molecular methods were presented.

## 2. Materials and Methods

### 2.1. Specimens

In total, the samples of sera and internal organs from 170 free-living brown hares were tested for the presence of EBHSV and RHDV2. Of these, 113 sera from wild brown hares shot off in three hunting districts located in Southeast and Central Poland between December 2020 and December 2021 were used for serological examinations. For the sake of clarity of the description, two neighbouring hunting districts of the Lublin Voivodeship (units no 199 and 216) are further treated as a common territory (Figure 1). Of this district, 103 samples of sera (batches I—IV) were collected over two successive hunting seasons from 2020–2021 (*n* = 73) and 2021–2022 (*n* = 30). The remaining 10 sera (batch V) were obtained in December 2021 from the Łódź Voivodeship (hunting unit 252). Moreover, five reference anti-EBHS hare sera were obtained in Poland in the period 1999–2000 (collection of the Department of Food and Environmental Virology of the NVRI) and two sera anti-RHDV2 taken from convalescent rabbits from RHD outbreaks in 2016 and 2021 (FMD laboratory’s own collection) were included for serological examinations.

### 2.2. The Characteristics of the Hunting Areas

The areas are suitable for hares. In hunting units No 199 and 216 (adjacent to each other), hare hunting on a limited scale has been practised continuously for many years. This region includes 13,600 ha of fertile soil with 25.2% of forest covering. It is considered that many small farms (<1 ha) and the strong heterogeneity of the environment are conducive to the maintenance and development of the hares’ population in this area, where in 2017, their number was estimated at 10 hares/100 ha [37]. The hunting district no 252, situated in Central Poland near the village Rozprza (51°18′09″ N; 19°38′44″ E), includes 9500 ha encompassing 49% of forests and 51% of arable agricultural land, with participation of grasslands, meadows and peat bogs. The collective hare hunt in the 2021/2022 hunting season was carried out for the first time after more than ten years of interruption. The decision was based on a noticeable increase in hare population density and the effectiveness of the predator reduction program implemented, including the red fox.

### 2.3. Hares Body Weight, Sex and Age

The body weight of the animals was determined with an accuracy of 0.1 kg immediately after the shooting. The sex of the hares was also determined directly in the field, immediately after weighing, on the basis of the inspection of secondary sex characteristics. The age of the hares was initially assessed on the basis of palpation detection of cartilage thickening on the outer part of the ulna, which occurs in young hares. This made it possible to divide the material into young (up to 1 year old) and adult (over 1 year old) hares [38,39]. Additionally, in order to verify the results obtained with the field method, the left eyeball was dissected from each hunted hare and preserved in 10% buffered formalin. In laboratory conditions, lenses were prepared from the secured material and dried in a laboratory dryer at a temperature of 100 °C for 24 h to a constant weight [40]. After drying, the lenses were weighed on a laboratory balance with an accuracy of 0.001 g and divided into those weighing up to 290 mg (juveniles) and those weighing over 290 g (adults) [37,40,41,42,43].

Additionally, 57 animals from seven voivodeships, collected over the period 2014–2021 from hare deceased due to traffic incidents (*n* = 22), found dead in forest and rural environments (*n* = 5), sent from the animal shelters (*n* = 8), and shot for hunting purposes (*n* = 22), were used for examinations (Table 1). The samples are archival biological material stored frozen (−20 °C). Almost all of them were collected and made available by another laboratory that had previously tested them for parasites and bacterial pathogens, but not for lagoviruses. Some of the samples (*n* = 8) came from young hares, about 2 months old, who died shortly after being placed in the animal shelter. These hares did not come from the breeding sites and were not used for reintroduction.

### 2.4. Serological Methods

Antibodies to EBHSV and anti-RHDV2 in sera from 113 hares were detected using two inhibition ELISA kits commercially available from the Istituto Zooprofilattico Sperimentale della Lombardia e dell’ Emilia (IZSLER), Brescia, Italy, in accordance with the manufacturer’s instructions. Each ELISA kit contains the appropriate negative and positive control sera, viral antigen, and specific Mabs conjugated to horseradish peroxidase, enabling highly sensitive detection of EBHSV or RHDV2 antibodies. The serum titre corresponds to the dilution of the serum tested, which inhibits the absorbance (492 nm) value of negative control serum by 50%.

### 2.5. Virological and Molecular Tests EBHSV and RHDV2 Detection

The internal organs liver (*n* = 167), lungs (*n* = 14), kidneys (*n* =13), spleen (*n* = 10), duodenum (*n* = 67), heart (*n* = 6), faeces (*n* = 3), and blood samples (*n* = 113) of 170 brown hares were tested for the presence of EBHSV and RHDV antigens using RHD-EBHS ELISA kit (IZSLER, Brescia, Italy) according to the manufacturer’s instruction, as previously described [44]. For the detection of the genetic material of the relevant lagoviruses, the real-time RT-qPCR with TaqMan primer and probe sets, which had been established for RHDV2 [21] and EBHSV [22] were used. Amplification was carried out using the QuantiTect Probe RT-PCR Kit (QIAGEN) and AB 7500 thermocycler.

## 3. Results

### 3.1. Seroprevalence

The total seroprevalence of EBHS virus antibodies in 113 free-living hares from two different geographical regions of Poland, collected from December 2020 to December 2021, has been estimated at 87.6% with a titre range of >10 ≤2560. In individual batches of the samples, the percentage of seropositive hares was 91%, 75%, and 90% in five lots tested. No EBHS antibodies were detected in 12.4% of the animals (Table 2).

The distribution of the EBHSV antibody in 113 sera showed 51.5%, 43.4%, and 5.1% positive results with titres ranging from 10–80, 160–640, and >640, respectively. The highest antibody titre (2560) was found in one adult female hare from Lublin voivodeship (batch III from hunting season 2020/2021). In the two groups tested in this province, the percentage of sera with low titres (10–80) was higher than sera with medium titres (160–640) and ranged from 61% to 33%. In one of the two remaining groups of the region, the percentage of hares with low and medium titres were evenly distributed (about 48%), while the other showed an advantage of medium titres over low titres in the ratio of 53% to 33%. In the hunting district located in the central part of Poland, the percentage of hares with low and medium titres ranged from 55.6% to 44.4%. The overall proportion of EBHS seropositive females (48.7%) was higher than that of males (38.9%) (Figure 2). The relationships in individual batches of animals were similar or the percentage of positive results for females was slightly higher than for males, both in the range of low and medium titres. Depending on age, a generally higher percentage of EBHS-positive animals was found in adult hares (50.4%) compared to young hares (37.2%). Similar relationships were observed in young and adult hares in five lots tested (Figure 3).

### 3.2. RHDV2 Serology

No RHDV2-specific antibodies were detected in the brown hares’ sera tested.

### 3.3. EBHSV and RHDV2 Detection

No EBHSV and RHDV2 antigens or their genetic material were detected in the internal organs and blood samples of 170 brown hares tested (killed for hunting purposes, found dead following traffic incidents in rural and forest areas, as well as in hares showing nonspecific symptoms that died in an animal shelter) using the RHDV-EBHSV CR ELISA kit and two RT-qPCR methods specific for each virus. RHDV2 and EBHSV virus RNA was not detected by real-time RT-PCR also in blood samples of seropositive EBHSV hares, including the hare with the highest anti-EBHSV antibody titres (2560).

## 4. Discussion

Data available from some European countries from the last 50 years show a drastic decline in the abundance of free-living brown (*Lepus europaeus*) and mountain (*Lepus timidus*) hares due to environmental, and demographic reasons, as well as the effect of diseases [1,2,45,46,47,48,49]. In Poland, a constant trend in the decline of the abundance of brown hares and grey partridges accelerated dramatically at the beginning of the 1990s [50,51]. At the end of the 20th century, the average autumn hare density decreased twice as compared to the 1970s and remained at a low level, ranging from 2 to 20 hares/100 ha, over the next two decades. Unfavourable trends in the density of the hares’ population influenced the safety of hunting exploitation of this species. As a result, in many hunting districts shooting was discontinued or suspended. In the years 2014–2017, only 13–18 thousand hares per year were obtained from hunting [50,52,53]. Among the main reasons for the decrease in the population of hares in Poland, apart from the increase in the number of natural predators, such as foxes, the chemicalisation of agricultural crops, and far-reaching urbanisation the emergence of the viral, deadly plague disease, EBHS, should be indicated [2,50]. The attempts to increase the native population of hares by introducing captive-bred animals bring limited results due to the low survival rate [54]. However, the latest official data show a slight, systematic increase in the total number of hares in Poland, from about 560,000 animals in 2010 to almost 710,000 in 2015 and 870,000 in 2021. The most hares were recorded in voivodeships of central and eastern parts of the country: Masovia, Lublin, and Łódź voivodeships [55]. Accordingly, serological studies on EBHS presented in this paper were carried out in hunting districts located in two of the aforementioned provinces, with the highest population density of hares (Figure 1). In terms of the structure of agricultural crops and intensification of agricultural production, the hunting areas from which samples for serological tests were obtained can be defined as areas of low of agricultural production intensity, with a predominance of small farms. These can be compared to one of the hunting districts in Central Poland, where recently, hare density from 16.4 to 37.5 hares/100 ha was found [45].

It can be assumed that EBHSV antibodies detected in this study in both juvenile (J) and adult (A) apparently healthy hares of both sexes were produced as a result of contact of these animals with EBHSV, despite the fact that the last time EBHSV-RNA was detected in Poland in 2012 [56]. No EBHSV-positivity in 57 hares from 2014–2021 (the specimens not related with hunting seasons 2020/2021 and 2021/2022 described) may be explained by the fact that only 8.8% (*n* = 5) of these samples comprised animals found dead in the field. As a highly pathogenic virus, EBHSV probably causes death in a short period so the window of opportunity to find positive hares is narrow.

On the other hand, since the first data on the occurrence of EBHS in Poland were described in the early 1990s, it can also be assumed that the disease has been endemic here since then [17,19]. The above assessment is reflected in the phylogenetic characteristics of the EBHS strains isolated in Poland between 1992–2004, confirming the close relationship of the EBHSV G104 strain from 2004 with younger European EBHSV strains [44]. New strains of EBHSV, which have recently been detected in the eastern regions of Germany adjacent to Poland, also point out the active circulation of EBHSV in this part of Europe. Moreover, genomic recombination between the non-structural part of the EBHSV genome and the genes of the RHDV2 structural proteins indicates new scenarios of genetic variation of both lagoviruses with the perspective of antigenic changes [33].

RHD has been endemic in Poland since 1988 [53,57]. The first classic strains of RHDV (currently classified in the genetic group GI.1c) were detected at the same time as in neighbouring countries—in Slovakia, Germany and the Czech Republic. The outbreaks were linked to the spread of the RHD pandemic wave, which began in China in 1984, then spread through Asia, and Europe and reached Mexico in North America [15]. At this point, it should be explained that the emergence and maintenance of RHD in Poland are related to the presence of numerous small-scale breeding rabbit farms, having a significant quantitative advantage over industrial breeding farms. In recent years, companion rabbits have gradually been playing an increasingly important role in the epidemiology of RHD. RHDVa strains appeared in Poland 7–8 years after their diagnosis in Italy and Germany [58,59,60,61]. In turn, the first RHDV2 strains were diagnosed in Poland in domestic rabbits (including companion animals) 6 years after their detection in France [25,33]. Currently, RHD outbreaks are still detected in domestic rabbit farms, caused by both RHDVa and RHDV2. However, the severity of the disease is not as high as in the early 1990s or the first decade of the 2000s. Contrary to many Western European countries, wild rabbits do not seem to play a significant role in the epidemiology of RHD in Poland. Their general low numbers and occurrence restricted to small enclaves may favour the preservation of the antigenic and genetic stability of RHDV [44]. So far, only the presence of RHDVa has been confirmed in these rabbits. As there were no wild rabbits in the study area and there was no direct evidence of the presence of RHDV2 in breeding rabbits in close proximity, at the time of the survey, the chance of detecting RHDV2 antigen or its RNA in hares was significantly lower than in a case of EBHSV.

NoRHDV2 transmission to the brown hares examined was also confirmed by the serological results. Since the pathogenic RHDVa and RHDV2 lagoviruses appeared in Poland a few years later than in many other European countries, it can be expected that in the near future, RHDV2 infections will also occur in native hares. To verify this assumption, it will be necessary to continue research and obtain samples from other regions of the country.

The results of the EBHS serological survey presented in this paper are similar to the data received in Italy in the early 1990s and previous serological investigations of Polish free-living brown hares from 1992–1995, which showed 95% and 86% seroconversion, respectively [13,19]. Italian data pointed out that the most numerous were the animals with low (≤160) to average (1:320–1:640) antibody titres [13]. A study by Frölich and colleagues performed in Poland in this period has shown a 38% prevalence of antibodies to EBHSV in hares in the Czempin area (Greater Poland Province) [20]. At that time, the high and mild seropositivity of EBHS in wild and captive hares was also recorded in Austria (87.5%), Slovakia (73–90%), and Germany (29%) [10,62,63]. The observations in areas where EBHS is endemic have shown that the circulation of the virus is linked to hare population densities, the age and sex of the animals. According to this, the young hares, less than 2–3 months old, are naturally resistant to EBHSV infection. In the light of these data, a high percentage of seropositive hares found in the presented research may result from a higher population density in the studied areas than it would appear from official data. In turn, early subclinical contact of juveniles with EBHSV would favour the development of long-term protective immunity and a high percentage of seropositive hares, which is also characteristic of areas with a high population density of hares and may explain their low mortality [36,64,65,66]. The results of our research indicate that in the analysed populations these key conditions, relating to a large percentage of seropositive young hares, were met (Figure 3). The serological study of brown hares performed in two independent territories located in Southeastern and Central Poland and used for hunting purposes confirmed the high seroprevalence of anti-EBHSV seroreagents and, thus, the circulation and impact of the virus on this species. The result of the presented serological study proves that EBHS is still endemic in Poland. Based on our results, it can be assumed that the endemic presence of EBHS, still one of the most important infectious diseases of hares, in addition to many other pathological factors [67] does not have such a negative impact on the decline in the population of these animals. At present, environmental changes are considered the main cause of a deep decline in brown hare populations [45]. Moreover, it can be concluded that, according to the data from other EBHS endemic infections [36], the high seroprevalence in the hares tested was the result of early and regular contact of young hares with EBHSV. This situation probably creates a certain state of equilibrium between the virus and the susceptible hares, finally leading to low animal mortality. It can also be assumed that hares living in freedom in the examined areas (possibly throughout the entire country) are prepared for contact with the current antigenic and genetic form of EBHSV.

## 5. Conclusions

The results of this study confirmed the maintenance of EBHSV in the field and its role as an important factor which can affect the mortality of the brown hare population in Poland. The high percentage of seropositive hares shows that EBHS still has a significant impact on hare density. This confirms the importance of diseases as an influencing factor responsible for the survival of small game, a key element in maintaining the balance and biodiversity in wildlife. The results obtained indirectly indicate the presence of a higher density level of hares than expected in the native population of free-living brown hares and the existence of a fairly high-level population immunity limiting the negative impact of the EBHSV. The lack of evidence for the presence of rabbit haemorrhagic disease virus 2 in the examined brown hares does not exclude the possibility of its appearance and supports the continuation of research on both pathogenic lagoviruses, taking into account the other areas of the existence of hares (a potential host of RHDV2) in the country.

## Figures and Tables

**Figure 1 viruses-14-02423-f001:**
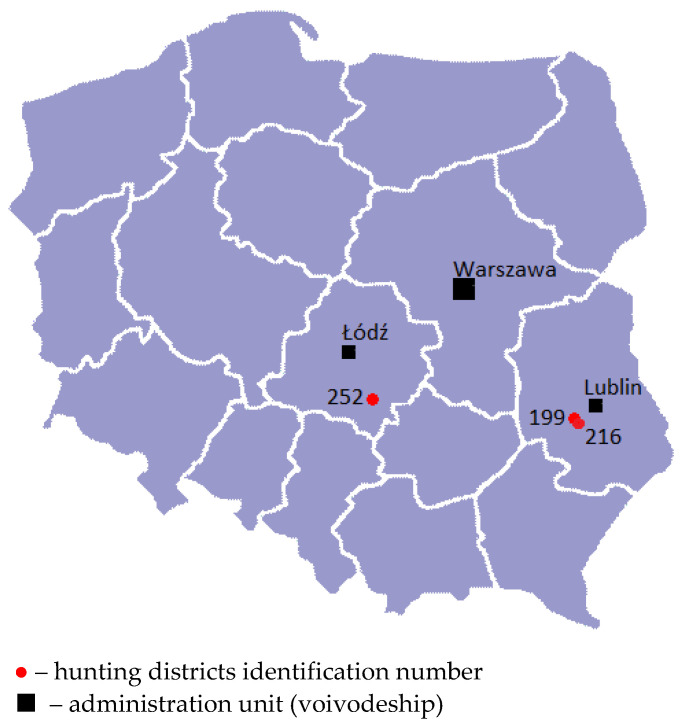
Location of the places of blood sampling used.

**Figure 2 viruses-14-02423-f002:**
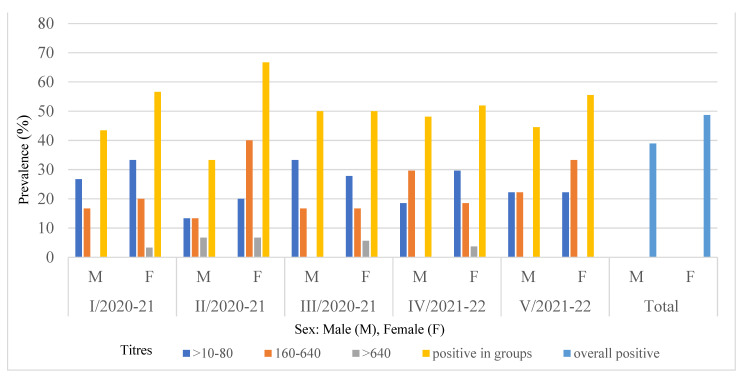
Distribution of EBHSV antibodies in seropositive hares, related to sex, place and period of sampling.

**Figure 3 viruses-14-02423-f003:**
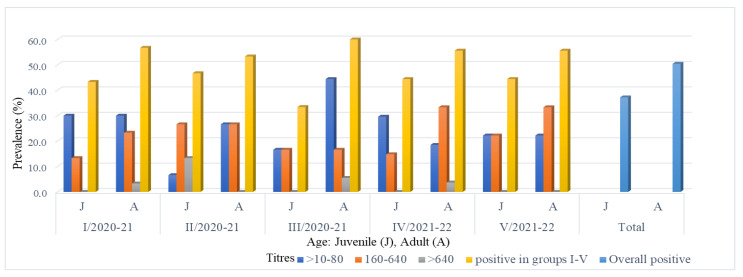
Distribution of EBHSV antibodies in seropositive hares, related to age, place and period of sampling.

**Table 1 viruses-14-02423-t001:** The type and origin of hare samples used in the study.

Date of Collection/Additional Information	Geographical Region of Poland /Voivodeship	No. of Hares	Sex M/ F	Weight	Specimens Collected						
			Age J/A	(kg)	Blood	Liver	Duod.	Lungs	Kidneys	Spleen	Faeces (f)/Heart (h)
Hunting season 2021–2022	Central/Łódź	10 serol. study batch V	2MJ, 2MA, 2FJ, 4FA	3.3–4.8	10	10	7	0	0	0	0
Hunting season 2021–2022	Southeastern/ Lublin	30 serol. study batch IV	5MJ, 9MA, 8FJ, 8FA	3.2–5.0	30	30	20	0	0	0	0
Hunting season 2020–2021		20 serol. study batch III	4MJ, 6MA, 4FJ, 6FA	3.5–5.0	20	20	5	0	0	0	0
		20 serol. study batch II	2MJ, 5MA, 7FJ, 6FA	3.0–5.0	20	20	10	0	0	0	0
		33 serol. study batch I	7MJ, 8MA, 6FJ, 12FA	3.6–5.0	33	32	32	0	0	0	0
2021/traffic incident	North-Central/ Kuyavian-Pomeranian (Bydgoszcz)	1	A	-	0	1	1	1	1	0	0
2021 /animals shelter		3	J (8 weeks)	-	0	3	3	0	0	0	0
2020 /animals shelter		1	J (4 weeks)	-	0	1	0	0	0	0	0
2020/traffic incident	Southeastern/ Lublin	1	A	-	0	1	0	1	1	1	0
2019/group hunting		5	A	-	0	5	0	0	0	0	1 (h)
2019/animals shelter	North-Central/ Kuyavian-Pomeranian (Bydgoszcz)	1	J (4 months)	-	0	1	1	0	1	1	0
2019/traffic incident		3	A	-	0	3	0	3	3	0	3 (h)
2018/animals shelter		3	J (10 weeks)	-	0	3	0	3	3	0	3 (f)
2016/traffic incident		1	J (11 weeks)	-	0	1	0	0	0	0	0
		2	A	-	0	2	0	2	2	0	0
2016/traffic incident	North/ Pomeranian (Gdańsk)	1	A	-	0	1	0	1	0	1	1 (h)
2016/found in the forest		1	A	-	0	1	0	0	1	0	1 (h)
2015/traffic incident	Southwestern/ Lower Silesia (Wrocław)	1	A	-	0	1	0	0	0	1	0
2015/found in the forest		1	A	-	0	1	0	1		1	0
2015/traffic incident	North-Central/ Kuyavian-Pomeranian (Bydgoszcz)	2	A	-	0	2	0	1	0	2	0
		8	A	-	0	8	0	0	0	2	0
2015/found in a rural environment		2	A	-	0	0	0	0	0	1	0
2015/group hunting	Central/Łódź	12	A	-	0	12	0	0	0	0	0
2014/found in a rural environmnet	Central/Masovia (Warszawa)	1	A	-	0	1	0	1	1	0	0
2014/traffic incident	Southwestern/Opole	2	A	-	0	2	0	0	0	0	0
2014/group hunting	Southeastern/ Lublin	5	A	-	0	5	0	0	0	0	0

A—adult (>1 year); J—juvenile (<1 year); M—male; F—female.

**Table 2 viruses-14-02423-t002:** Prevalence of antibodies to EBHSV in brown hares (*Lepus europaeus*) in the southeastern and central region of Poland over the period December 2020—December 2021.

Batch/ Hunting Season	No. of Hares Tested	Positive								Negative	
		No.	Percentage ^1^	Anti-EBHSV Antibody Titre Distribution ^2^ Titre (Pab ELISA/Mab ELISA)						No.	Percentage ^1^ %
				>10–80		160–640		>640		<10	
				No.	%	No.	%	No.	%	No.	%
I/2020–2021	33	30	90.9	18	60	11	36.66	1	3.33	3	9.1
II/2020–2021	20	15	75	5	33.33	8	53.33	2	13.33	5	25
III/2020–2021	20	18	90	11	61.1	6	33.33	1	5.6	2	10
IV/2021–2022	30	27	90	13	48.1	13	48.1	1	3.7	3	10
V/2021–2022	10	9	90	4	44.4	5	55.6	0	0	0	10
Total	113	99	87.6	51	51.5	43	43.4	5	5.1	14	12.4
Males	49	44	38.9	23	23.2	20	20.2	1	1	5	4.4
J	20	19	16.8	10	10.1	8	8.1	1	1	1	0.9
A	29	25	22.1	13	13.1	12	12.1	0	0	4	3.5
Females	64	55	48.7	28	28.3	23	23.2	4	4	9	8
J	28	23	20.4	13	13.1	9	9.1	1	1	5	4.44
A	36	32	28.3	15	15.2	14	14.1	3	3	4	3.5

^1^ Percentage of positive/negative in relation to the number of hares tested (in groups/total); ^2^ percentage of positive sera with the specified titre in relation to the total number of positives. A: adult (>1 year); J: juvenile (<1 year).

## Data Availability

Not applicable.

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
