# Peer review of "European Brown Hare Syndrome in Poland: Current Epidemiological Situation"

_viruses, 2022, doi:10.3390/v14112423_

Round 1

Reviewer 1 Report

The manuscript entitled “European Brown Hare Syndrome in Poland: Current Epidemiological Situation” by Andrzej Fitzner and collaborators (Viruses, 1955209) describes a serological survey conducted in European brown hares from three districts of Poland. The aim was to evaluate the current epidemiological situation of two lagoviruses based on the detection of EBHSV and RHDV2 antibodies in 113 sera collected from free-ranging hares harvested in central and southeastern Poland in the 2020-2021 and 2021-2022 hutting seasons. The results of the serological examination showed that nearly 88% of tested hares were positive for EBHSV antibodies. No RHDV2 antibodies were detected.

Additionally, a smaller virological survey was conducted in 57 hares sampled between 2014 and 2021 to investigate the presence of EBHSV and RHDV2 RNA (or antigens) in blood and organs. No positives were detected.

The authors concluded that EBHSV is circulating among the hare populations in Poland being maintained in the field and playing a role in balancing the populations survival and abundances.

 Although the study is interesting and deserves being published, the current version of the manuscript requires a careful revision regarding minor errors and faults. Also, the manuscript language will benefit from being read and amended by an English native speaker.

 Specific comments.

Title

To keep the title as it is currently (European Brown Hare Syndrome in Poland: Current Epidemiological Situation), since the study refers to three specific provinces of Poland, a sentence has to be added to the manuscript (Discussion section) saying that the findings can probably reflect the reality of the whole country.

Abstract

Line 15: Add the species names of Brown hare (Lepus europaeus) and Mountain hare (Lepus timidus).

Line 16. The highly lethal -> this highly lethal

Line 19. Rabbit haemorrhagic disease virus 2 (RHDV2)

Introduction

Line 47: presence of EBHSV specific antibodies

Line 48: as well as virological tests

Line 50: genetic material-> RNA

Line 51- 53: Please rephrase the sentence and explain better what you want to say by “indicating the possibility of passive transmission of EBHSV via other than secretions and excretions biological material of diseased hares [24].” Maybe by adding that sympatric hare’ predators may play a role in the passive dissemination of the EBHSV.

Line 57-60: RHDV2 jumped several species barriers including non-lagomorph species such as the Eurasian badger (Abade dos Santos et al, 2021).

Line 64: please explain what you mean by “two serotypes of RHD”.

Materials and methods

Line 93: Number abbreviation is “No.”

Line 95 and 100: hundreds uses comma not dot (13,600 ha). Please apply the same criteria to the entire manuscript (lines 205 and 205).

Line 121-124: Information is missing in this sentence. Additionally, animals from seven voivodeships, collected over the period 2014-2021 from hare deceased due to the traffic incidents (n=22), found dead in forest and rural environment (n=5), sent from the animal shelters (n=8), and shot for hunting purposes (n=22), were implemented for examinations (Table 1).

One cannot guarantee that an animal that suffers a traffic incident is healthy as it is common to see sick animals in roads and near populations. What do you mean by “sent from the animal shelters”? released into nature?.

“were implemented” does not make sense.

Line 137: please use the same criteria in the sentence (numbers between brackets).

Line 141: lagoviruses

Line 141: “real-time qRT-PCR assays” is incorrect. q=quantitative=real time. Please replace by RT-qPCR. Since the R means reaction/reactions, please erase “assays” as reactions assays is a repetition.

Line 144: the apparatus and software used is missing here.

Results

Line 171: The results obtained for the RHDV2-RNA investigations should be discussed having in mind the prevalence of RHDV2 in the wild rabbit populations in the studied areas.

Discussion:

Line 180: The Iberian hare (Lepus granatensis) also suffered an abrupt decrease after the emergence of ha-MYXV in Spain and Portugal in 2018. I think this information would show the reader that most of hare species are decreasing which is alarming for biodiversity. 

Line 218: since lagoviruses are not isolable, please replace “last virus isolate was identified in Poland in 2012” by “the last time EBHSV-RNA was detected in Poland was in 2012”.

In this section, it should also be discussed why such a high seroprevalence of near 88% was not accompanied by EBHSV-positivity in the 57 tested specimens. This may be explained by the fact that only 8.8% (n=5) of this sample comprised animals found dead in the field. As a high pathogenic virus, EBHSV probably causes dead in a short period so the window of opportunity to find positive hares is narrow.

Figures

 Figure 1. In my opinion, it would be worthwhile to add the number of samples obtained from each region to figure 1. In the legend please add the meaning of the black squares and red dots.

 Tables

Table 1. Why are Faeces and Heart in the same column? Why is “traffic incident” in small letters and  “Animals shelter” in capitals?

 Table 2. please replace commas by dots.

Author Response

Thank you very much for a thorough revision of the manuscript, your valuable comments and consideration of this manuscript for further revision. We have carefully revised the manuscript according to helpful Reviewers suggestions. All changes and responses to the comments were listed in the revision note.

            We appreciate your continued evaluation of this manuscript.

Yours sincerely,

Andrzej Fitzner

On behalf of all Authors

Reviewer 2 Report

see the encolsed file. 

Author Response

(The authors gave the same response as above.)

Reviewer 3 Report

This article describes the current epidemiological situation regarding EBHS in parts of Poland. As such, this update it is of interest to the readers of Viruses.

In its current presentation, the article needs major revisions regarding data presentation, unity between introduction and discussion, contents of discussion.

General

Replace “voivode” for the English translation in the whole manuscript.

RHDV antigens are studied, but lacks any discussion.

Abstract

l.22: mention here whether the hares used for serology were dead or alive.

l.25: indicate wether the used hares were ill, shot, free-ranging or in captivity.

Introduction

L 70: virological and molecular studies were carried out on which animals? Please add.

l.73: same individuals?

Material and Methods

l.77- 89: very confusing data presentation. On geographical reference, consider re-numbering of hunting units 199 (etc) into simple numbers e.g. 1-5, or a-e. I do not get an idea of which tissue from which animal is used for which purpose, as animal numbers, geographical information and used samples are a knot in this description.

Fig. 1: numbers and text are too small to read.

l.93 and 108: use as subtitles

l 98-107: too long, rephrase and summarise to max. 5 lines. Basis message: the areas are suitable for hares, in one area hunting was continuously present, while in the other, during a 10-year period, it was absent. Please elaborate on this difference in hunting history the discussion.

Table 1. Fair start, but confusing in present form.

Split column 1 in two parts: year and additional information.

Define the dates of the hunting season in the caption.

Relate names of geographical location to a geographical map. The general public of this journal likely is not aware of the details of Polish geography.

Why combine sex and age in one column?

L 136:  omit “for …”, as this is not done in line 127 too.

Results

L 154: why this choice? Why do we want to know this? Please mention this somewhere in MM, or in discussion, or omit these numbers when not elaborated in discussion.

L 157: Please specify distribution.

L 171: Same number of animals tested as for EBHS? Same individuals? Were any individuals tested for both EBHS and RHDV?

Discussion

Discussion of RHDV results lacks completely. Please elaborate on this issue.

Share some thoughts regarding representation of used geographical distribution for Polish situation in general.

L 181-201: too extensive, please reduce drastically. The summary is already nicely presented in lines       178-180.

l. 202 -215: Too long, and inconsistent use of density quantities: hares in Poland as a country, elsewhere hare per square kilometer.

L 216-220: do you try to suggest that your findings of EHBS antibodies in likely sound hares indicate/ suggest endemic circulation of the virus? Then put it forward more clearly.

L 224-228: interesting information, but please do try to relate more clearly to your own findings.

L 232: most representative for what?

L 238 – 244: please relate to your own findings

L 244-248: does not seem to add something essential to this article. Consider to omit completely, or relate more clearly to your own findings.

L 251: your data do suggest endemic circulation of EBHS in these hares. Specify the mentioned impact on the species. Now, you seem to suggest that EBHS also has a greaty contributes to population decline, but maybe you try to say the opposite. Please specify your conclusion here.

In my opinion, negative impact of EBHS when endemic, is debatable because of your own findings. To state anything regarding the impact of a virus, you also need concurrent pathology data of spontaneously deceased hares.  These are missing. It might help te add relevant data here. Generally, endemic viral circulation is much less harmful at population level, than introduction into a naïve population, high mortality and few survivors. Please elaborate on these issues. Quantify (formatively) potential effect of EBHS on hare-population decrease compared to the other mentioned factors.

L 258-260: What do you try to state? What is the relation to your findings?

Conclusion:

L 265: abundance = density

Add interpretation of RHD findings

Author Response

(The authors gave the same response as above.)
